# Adrenomedullin Therapy in Moderate to Severe COVID-19

**DOI:** 10.3390/biomedicines10030533

**Published:** 2022-02-24

**Authors:** Toshihiro Kita, Kazuo Kitamura

**Affiliations:** Department of Project Research, Frontier Science Research Center, University of Miyazaki, Miyazaki 889-1692, Japan; kazuokit@med.miyazaki-u.ac.jp

**Keywords:** adrenomedullin, translational study, clinical trial, COVID-19, pneumonia, acute respiratory distress syndrome

## Abstract

The 2019 coronavirus (COVID-19) pandemic is still in progress, and a significant number of patients have presented with severe illness. Recently introduced vaccines, antiviral medicines, and antibody formulations can suppress COVID-19 symptoms and decrease the number of patients exhibiting severe disease. However, complete avoidance of severe COVID-19 has not been achieved, and more importantly, there are insufficient methods to treat it. Adrenomedullin (AM) is an endogenous peptide that maintains vascular tone and endothelial barrier function. The AM plasma level is markedly increased during severe inflammatory disorders, such as sepsis, pneumonia, and COVID-19, and is associated with the severity of inflammation and its prognosis. In this study, exogenous AM administration reduced inflammation and related organ damage in rodent models. The results of this study strongly suggest that AM could be an alternative therapy in severe inflammation disorders, including COVID-19. We have previously developed an AM formulation to treat inflammatory bowel disease and are currently conducting an investigator-initiated phase 2a trial for moderate to severe COVID-19 using the same formulation. This review presents the basal AM information and the most recent translational AM/COVID-19 study.

## 1. Introduction

The novel coronavirus disease (COVID-19) pandemic is sweeping the globe, and it is impossible to foresee how it might end. Over five million patients have died before the end of 2021 [1]. Recently introduced vaccines, oral antiviral medicines, and antibody formulations can suppress COVID-19 symptoms and decrease the number of patients with severe conditions [2,3,4,5,6]. However, complete avoidance of severe conditions has not been achieved, and there are insufficient treatment methods for severe conditions [7]. For example, neutralizing monoclonal antibody (LY-CoV555) plus remdesivir was found to be ineffective in hospitalized patients with COVID-19 [8]. Therefore, it is still critical to determine how to save the lives of patients with severe conditions. In particular, patients who require mechanical ventilation for exacerbated pneumonia due to COVID-19 have a high mortality rate [9,10]. Severe pneumonia cannot be treated with antiviral drugs only, and alternative approaches are needed. Innovative drugs to alleviate tissue damage in severe COVID-19 cases using a different approach from conventional antiviral drugs are being investigated.

Adrenomedullin (AM) is a vasodilatory bioactive peptide that exhibits anti-inflammatory and tissue-protective actions. AM plasma concentration has been shown to increase remarkably in patients with sepsis and severe pneumonia. AM acts as an endogenous defensive substance [11,12,13,14,15,16,17,18,19,20,21,22,23]. Exogenous administration of AM has been shown to improve organ damage caused by sepsis and pneumonia in experimental animals [24]. Additionally, AM prevented ventilator-induced lung injury (VILI) in a mouse pneumonia model [25]. Mechanical ventilation worsened lung injury, which then caused multi-organ failure, including the liver, kidneys, and intestinal tract. However, AM administration in mice considerably decreased lung injury and suppressed liver and intestinal tract disorders [25]. AM decreases tissue injury by improving cytokine control and the barrier function of endothelial cells. Contrarily, AM plasma concentration and related proAM fragments were shown to increase in patients with COVID-19, and more importantly, the increase in AM was closely related to the prognosis of patients with COVID-19 [26,27,28,29,30,31,32,33,34,35,36]. These data suggest that AM plays an important role in COVID-19 pathophysiology; thus, it could be a candidate alternative treatment for COVID-19. Therefore, we investigated whether AM could reduce organ damage in patients with severe pneumonia caused by COVID-19 and whether it could improve patient prognosis. Figure 1 illustrates the current therapeutic drugs for treating COVID-19 in Japan. According to Japanese guidelines, patients grouped in the moderate 1 category have pneumonia or dyspnea but do not need oxygen support, while those in the moderate 2 category require supplemental oxygen. Patients classified as having the severe form of the disease receive ventilatory support and/or are admitted to the intensive care unit (ICU). Major therapeutics in the early and mild phases of COVID-19 are sufficient, but there are limited or only partially effective treatments in the advanced phase. AM is a promising candidate for COVID-19; however, it is currently uncertain whether it plays a significant role in the advanced state of the disease.

In this review, we present the anti-inflammatory and tissue-protective effects of AM and the relationship between AM and COVID-19. We then introduce two currently ongoing investigator-initiated clinical trials using AM for moderate to severe pneumonia caused by COVID-19 (https://jrct.niph.go.jp/latest-detail/jRCT2071200041 (accessed on 9 October 2020) and https://jrct.niph.go.jp/latest-detail/jRCT2071210038 (accessed on 23 June 2021)).

## 2. Biosynthesis of AM and Its Receptors

AM, a 52-amino acid peptide, has a ring structure with a disulfide bond and an amidated C-terminal Tyr [37]. The ring structure and the amidated C-terminus are essential for bioactivity. Calcitonin (CT), the α- and β-calcitonin gene-related peptide (CGRP), amylin and adrenomedullin 2 (also known as intermedin) share structural features, namely, an amidated C-terminus and an N-terminal disulfide bond [38,39]. Therefore, these peptides, including AM, belong to the CT/CGRP superfamily [38,39]. Figure 2 illustrates the AM synthesis process. First, a precursor consisting of a 185-amino acid peptide, designated preproAM, is synthesized and then processed into proadrenomedullin (proAM). ProAM is further processed into four segments: proAM N-terminal 20 peptide (PAMP)-Gly, mid-regional proadrenomedullin (MR-proADM), AM-Gly, and C-terminal proAM [37]. Initially processed PAMP and AM contain an additional C-terminal Gly, and these intermediate forms are biologically inactive. The intermediate forms of the peptides are enzymatically converted into the amidated C-terminal, resulting in the mature bioactive forms; however, a limited part of the peptides is converted [40]. MR-proADM is biologically inactive but stable in the bloodstream, and thus, MR-proADM can serve as a useful biomarker for AM synthesis [41].

The CT/CGRP family, including AM, act on G protein-coupled receptors (GPCRs), namely the calcitonin receptor (CTR), and calcitonin receptor-like receptor (CLR). CTR can function on its own, but CLR requires the presence of chaperone molecules, called receptor activity-modifying proteins (RAMPs). AM and CGRP receptors are composed of CLR and three RAMP forms (1, 2, and 3). CLR with RAMP1 yields CGRP receptor, CLR with RAMP2 yields the functional AM receptor (AM1 receptor), and CLR with RAMP3 yields another functional AM receptor (AM2 receptor) [42]. The crucial role of AM and the AM1 receptor axis for vascular homeostasis has been clarified in experiments using knockout (KO) models, where AM KO mice developed lethal vascular abnormalities at mid-gestation resulting in extreme hydrops [43,44]. Interestingly, CGRP KO mice were normal, and only RAMP2 KO caused lethal abnormalities similar to those in the AM KO mice [45,46]. Moreover, the AM2 receptor is important in lymph vessel function [47].

## 3. AM and Sepsis

Several studies have provided useful information on AM in sepsis, one of the most severe infection states; the results suggest that AM is closely related to pathophysiology of sepsis and septic shock [24]. AM and MR-proADM plasma concentrations can markedly increase in patients with sepsis and septic shock, depending on the severity of the disease. More importantly, increased AM or MR-proADM levels are useful predictors in sepsis and septic shock prognosis [11,12,13,14,15,16,17,18,19,20,21,22,23]. A meta-analysis has shown that AM and MR-proADM exhibit high sensitivity and specificity as prognostic sepsis markers with 95% confidence interval values of 0.83 (95% CI: 0.79–0.87) and 0.90 (95% CI: 0.83–0.94), respectively [48]. The close relationship between AM and sepsis suggests that AM is a crucial factor in sepsis. 

AM and AM receptors are ubiquitously expressed in the body, including the heart, lungs, kidneys, adrenal glands, and intestines [49]. In particular, high AM expression has been confirmed in the vasculature, in which it is expressed in the endothelium and vascular smooth muscle cells [50]. The main function of AM in the vasculature is the maintenance of vascular tone and integrity. AM is one of the strongest vasodilators among the vasoactive peptides [51] and maintains vascular tone through direct action on vascular smooth muscle cells and nitric oxide produced in the endothelium [52,53]. More importantly, it directly stabilizes endothelial barrier function and suppresses further pro-inflammatory impairment of the vessel wall [54]. Sepsis and subsequent septic shock cause excessive vasodilation and breakdown of vascular integrity, contributing to organ damage and mortality. The early sepsis hyperdynamic phase progresses to the late hypodynamic state, causing organ damage and mortality. AM is crucial in causing the hyperdynamic phase but delays the transition to the hypodynamic state [55], the key point of the organ-protective effect of AM, even though it is a vasodilative peptide. Contrarily, vascular endothelial dysfunction is an important process in tissue inflammation, in which the loss of endothelial barrier integrity results in extravasation of fluids and molecules, causing edema and inflammation, and finally, tissue dysfunction [56,57].

Table 1 summarizes the selective effects of AM on experimental sepsis and septic shock. Genetic intervention AM models have demonstrated that AM is an endogenous protective factor against sepsis [58,59,60]. Exogenous AM administration can prevent this process and protect organs in experimental sepsis and septic shock models [25,61,62,63,64,65,66,67,68]. Although endogenous AM levels are increased in sepsis, the additional administration of exogenous AM prevents organ damage and improves the survival rate in animal models [25,61,62,63,64,65,66,67,68]. In particular, AM significantly decreases vascular, alveolar, and intestinal epithelial permeability, an essential factor in organ protection [25,62,63,67]. Based on the reported experimental models, Figure 3 illustrates a simplified cascade of AM functions as an organ-protective factor against sepsis. Although AM is induced by bacterial endotoxin as an endogenous protective factor [69], the quantity produced is insufficient to alleviate the disease. Therefore, exogenous AM administration can be beneficial in sepsis. Unfortunately, its effects have not yet been studied and approved for patients with sepsis, and its effects on patients are unknown.

## 4. Adrecizumab and Sepsis

Adrecizumab is a non-neutralizing humanized high-affinity antibody directed against the AM N-terminus. Adrecizumab binds to AM to form a large molecule; this modification protects the bound AM from proteolytic enzymes and results in a longer AM half-life in the bloodstream. The plasma concentration of AM in humans markedly increases and lasts for a long time after adrecizumab administration [70]. Interestingly, the MR-proADM plasma concentration has not been shown to increase with adrecizumab administration; therefore, AM biosynthesis was not shown to increase by adrecizumab [70]. More importantly, the AM N-terminus is unrelated to its bioactivity. Therefore, the adrecizumab antibody is expected to enhance the beneficial effects of endogenous AM in patients with high AM plasma levels. Adrecizumab reduces vascular leakage and organ dysfunction and improves survival in several sepsis models [71,72,73]. Based on these data, a phase 2a clinical trial of adrecizumab in septic shock patients was conducted [74]. Unfortunately, this trial did not demonstrate a definitive benefit of adrecizumab in septic shock [74]. The mechanisms of septic shock are complex, and it is a difficult target for clinical trials. For example, a meta-analysis has shown that the clinical relevance of a metabolic resuscitation cocktail (thiamine, ascorbic acid, and hydrocortisone) for sepsis is questionable [75]. Alterations to the trial protocol for adrecizumab may be needed in future trials.

## 5. Overview of Therapies for COVID-19

Great efforts in the development of therapeutic drugs against severe acute respiratory syndrome coronavirus 2 (SARS-CoV-2) have been concentrated after the beginning of the COVID-19 pandemic. To start, vaccination is the most effective way to control pandemic diseases. In particular, the mRNA-based vaccine, BNT16b2, produced by Pfizer Inc. and BioNTech SE was 95% effective in preventing COVID-19 [76]. Additionally, BNT16b2 was 94% effective for preventing hospitalization and 92% effective for preventing severe disease [77]. We believed that the mRNA vaccine would end the COVID-19 pandemic. However, this hope was dashed by the emergence of SARS-CoV-2 variants, such as Delta [78] and the recent Omicron variant [79]. It has become clear that a vaccine alone cannot end COVID-19. A second method of mitigating disease is antibody formulations against SARS-CoV-2. The neutralizing monoclonal-antibody combination agent (bamlanivimab plus etesevimab) reduced 70% of COVID-19–related hospitalizations or deaths [5]. However, this combination was less effective for the Gamma variant [80], and thus antibody formulations have the same problem that vaccines do for controlling variants of SARS-CoV-2. The most recent developments are oral antiviral drugs for SARS-CoV-2 [3,4]. Pfizer’s paxlovid, with 89% effectiveness, especially, may be a game changer in the treatment of COVID-19. Another anti-viral pill, molnupiravir, is also useful for patients with COVID-19, but this pill should be administered within 5 days of the onset of symptoms [81]. Remdesivir is the first approved antiviral agent, and is being used as a standard care drug for mild to severe disease caused by COVID-19. However, the benefit of remdesivir may be questionable [8], and the World Health Organization (WHO) recommendation guidelines for its use are weak or conditional [82].

Vaccines, antibody formulations, and orally bioavailable antiviral drugs are all generally useful for preventing severe illness caused by COVID-19; however, a certain proportion of patients with COVID-19 will develop severe disease, where these agents do not aid in recovery. The immune system overreaction is crucial in the progression of the acute pneumonia and resulting multiorgan damages in severe COVID-19. Therefore, immune suppression, such as that provided by corticosteroids, is effective in advanced stages of COVID-19. However, at the same time, avoidance of superimposed infection is also important with immunosuppressant therapy [83]. Significant decreases in the mortality rates of hospitalized patients with COVID-19, most of whom had moderate to severe pneumonia, were confirmed in well-organized clinical trials with dexamethasone or tocilizumab [84,85]. The benefit was clear but the differences in death at 28 days between active treatment vs. standard care were small (dexamethasone; 22.9% vs. 25.7%, rate of risk ratio 0.83, tocilizumab; 31% vs. 35%, rate of risk ratio 0.85) [84,85]. The benefit of baricitinib for hospitalized COVID-19 patients was also confirmed in a randomized, double-blind clinical trial [86]. This trial included patients with mild to moderate COVID-19, and the composite primary endpoint was the proportion who progressed to high-flow oxygen, non-invasive ventilation, invasive mechanical ventilation, or death by day 28. The difference between baricitinib and placebo was also small: 27.8% vs. 30.5% (odds ratio 0.85) [86]. The superiority of combination therapy, namely, baricitinib plus remdesivir compared with remdesivir alone, for hospitalized COVID-19 patients was also reported [87]. This trial enrolled patients with moderate (68.3%) and severe (31.7%) illness where everyone was administered remdesivir and additionally received baricitinib or placebo. The primary outcome measure was the time to recovery during the 20 days after enrollment. The median time to recovery (95% CI) was 7 days (6–8) in the baricitinib group and 8 days (7–9) in the placebo group (rate ratio 1.16, *p* = 0.03) [87]. The mortality rate over the entire trial period was 5.1% in the baricitinib group and 7.8% in the placebo group, and the hazard ratio for death was 0.65 (95% CI, 0.39 to 1.09) [87]. These trials have demonstrated significant but partial benefits of immune modulators for moderate to severe pneumonia caused by COVID-19. 

Abnormalities of coagulation and disseminated intravascular coagulation (DIC) are noticeable characteristics of COVID-19 and anti-coagulation therapy is crucial for the treatment of advanced stages of COVID-19. D-dimer is the representative marker for coagulation abnormalities and thus the marker for the prognosis of patients with advanced COVID-19 [88]. However, advanced COVID-19 is a complicated disease and a single marker, such as D-dimer, to detect coagulation abnormalities is insufficient [89]. DIC associated with COVID-19 is very different from that of septic DIC, and both thrombotic and hemorrhagic pathologies should be noted [90]. Thrombosis treatment is essential for COVID-19, in addition to antiviral and cytokine storm treatments. Initial anticoagulant treatment with low molecular weight heparin has been shown to reduce mortality by 48% at 7 days and 37% at 28 days, which demonstrates the major impact anticoagulation therapy can have [88]. Therefore, antithrombotic prophylaxis is strongly recommended for all hospitalized patients with COVID-19, but therapeutic anticoagulation is adapted for carefully selected patients to avoid severe hemorrhagic complications [91].

## 6. AM and COVID-19

SARS-CoV-2 infects the host via the angiotensin-converting enzyme 2 (ACE2) receptor [92]. The ACE2 receptor is not only widely expressed in the lungs but also in the heart, kidneys, and endothelial cells [93]. Therefore, SARS-CoV-2 causes endotheliitis, resulting in vascular dysfunction and subsequent tissue damage [94]. Vascular dysfunction comprises impaired vascular blood flow, coagulation, and leakage and induces organ dysfunction and edema [95,96]; it is a key factor in COVID-19 pathology [97]. AM reduces vascular hyperpermeability and promotes endothelial stability and integrity during severe inflammation [54]. Several mechanisms for the stabilization of the endothelial barrier by AM have been reported. For example, AM regulated the actin–myosin cytoskeleton [98] and prevented stress fiber formation through a cAMP-dependent mechanism [99]. AM also diminished hydrogen peroxide-induced edema development in isolated perfused rabbit lungs through a cAMP-driven mechanism [99]. Therefore, it is expected to be a therapeutic agent against COVID-19-induced endotheliitis [100]. MR-proADM, an AM synthesis marker, has been recognized as the best marker for sensitivity and specificity in sepsis endothelial dysfunction [101]. A significant MR-proADM increase has been reported in patients with COVID-19, especially in severe conditions, such as acute respiratory distress syndrome (ARDS) [26,27,28,29,30,31,32,33,34,35,36]. More importantly, increased MR-proADM is the best predictor of mortality in patients with COVID-19; the area under the receiver operating characteristic curve (ROC AUC) was distributed in a high range (0.78–0.951) [27,28,29,31,32,33,34,35]. MR-proADM plasma concentrations in COVID-19 survivors did not increase after hospital admission; however, concentrations increased significantly in non-survivors [35]. The ROC AUC of MR-pro-ADM increased by 0.78 within 24 h of admission to 0.92 on days 5–6 of hospitalization [33]. In addition to plasma concentrations, AM RNA expression was higher in patients with COVID-19 than in patients with other respiratory infections [102], and its expression is associated with the severity of COVID-19 [102]. Significant increases in the active form of AM, denominated bioactive ADM, have also been reported [103]. Active AM reflects real-time dynamics of AM secretion; thus, a marked increase in active AM in COVID-19 is important. These findings demonstrate the close relationship between AM and COVID-19; however, there are no data concerning the effects of exogenous AM administration on SARS-CoV-2 infection, including in experimental animals. Only exploratory research using adrecizumab for critical patients with COVID-19 has been reported [104] from a single-arm open study with no controls, conducted at a single-center, in which eight patients with critical illnesses, such as ARDS, caused by COVID-19 received the anti-AM antibody, adrecizumab. AM plasma concentration was markedly increased after adrecizumab administration, and only one patient died [104]. This 12.5% mortality rate (1 of 8) was lower than the 42% mortality rate (22 of 53) in similar patients at the hospital during the same period [104]. The results are promising; however, further verification studying a larger cohort with a control group is required.

The immune system overreaction, called a cytokine storm, is essential for progression of pneumonia to a severe state in COVID-19. Experimental studies demonstrated that exogenous AM administration into several septic models caused suppression of proinflammatory cytokines, such as tumor necrosis factor (TNF)-α or interleukin (IL)-1β [63,65,68]. Therefore, the suppressive effect of AM on proinflammatory cytokines in patients with COVID-19 may be expected; however, that seems to be uncertain. The suppressive effect of AM on interferon-γ and TNF-α in an experimental model of colitis was also confirmed [105]. However, the decreases in plasma concentration of IL-6 and TNF-α were quite limited in patients with ulcerative colitis (UC) by administration of AM [106]. Tissue cytokine changes may not always reflect the plasma concentration of cytokines. Consequently, we must evaluate the effects of AM on cytokines in future trials.

## 7. Clinical Trials Using AM for COVID-19

### 7.1. Progress of Clinical Trials Using AM

AM was discovered in 1993, and the first human study was conducted in 1997 [107]. Early studies focused on the vasodilative effect of AM and used a relatively high dose to alter hemodynamics and humoral factors [108]. However, controlling the vasodilative effect of AM in general use is difficult; therefore, AM has not been adopted as a therapeutic agent for cardiovascular diseases [108]. The turning point was discovering the mucosal healing effect of AM in experimental colitis [109]. Human AM studies to detect the safety range of AM for patients started in 2010 [110,111]. The beneficial effects of AM in UC and Crohn’s disease (CD) have been confirmed in exploratory studies [106,112,113], in which the AM dose was set to the minimum to avoid hypotension and for patient safety. An 8 h per day intermittent AM administration protocol was used to avoid an unwanted blood pressure decrease at midnight. Based on these studies, the diverse effects of AM, including organ protection, anti-inflammatory effects, and tissue repair, were confirmed within the AM dosage range, without uncontrollable effects on hemodynamics. We developed an AM formulation according to good manufacturing practice and confirmed its safety and tolerability up to a dose of 15 ng/kg/min for healthy males in a phase 1 investigator-initiated clinical trial [114]. The beneficial effects of AM in patients with UC were confirmed in a phase 2a investigator-initiated, double-blind, placebo-controlled, multicenter clinical trial [115]. In this trial, we determined three AM doses (5, 10, and 15 ng/kg/min); however, only 15 ng/kg/min was effective in patients with UC [115]. Additionally, valuable but limited effects of AM were confirmed in phase 2a investigator-initiated clinical trials for biologic-resistant CD (unpublished data).

### 7.2. Phase 2a Clinical Trial for COVID-19

After the COVID-19 pandemic, we started an investigator-initiated clinical trial for COVID-19. Initially, we focused on patients with severe pneumonia who required mechanical ventilation because an alternative resolution is frequently required for this critical condition. Müller-Redetzky et al. [25] suggested that AM is effective in ventilator-associated organ damage, such as VILI in a rodent model. This study is an investigator-initiated phase 2a, randomized, double-blind, multicenter, placebo-controlled clinical trial (https://jrct.niph.go.jp/latest-detail/jRCT2071200041 (accessed on 9 October 2020)). The inclusion criterion was patients who required mechanical ventilation for respiratory failure caused by COVID-19. The primary endpoint is the duration of mechanical ventilation. Major secondary endpoints include clinical six-category ordinary scores until 30 days after the start of the test drug. The categories are: (1) discharged or ready for discharge; (2) hospitalized, not requiring supplemental oxygen; (3) hospitalized, requiring supplemental oxygen; (4) hospitalized, receiving noninvasive ventilation or use of high-flow oxygen devices; (5) hospitalized, receiving invasive mechanical ventilation or extracorporeal membrane oxygenation; and (6) death. Within 24 h from the start of mechanical ventilation, continuous intravenous AM infusion was administered for 72 h, followed by intermittent administration for 8 h per day until weaning from mechanical ventilation (Figure 4A). We selected the highest dose of AM (15 ng/kg/min) and continuous infusion for 72 h at the beginning of the treatment because the critical stage of COVID-19 should be treated with the best care available. This dose of AM has been developed for therapy of inflammatory bowel disease. The increase in plasma concentration of bioactive AM was limited in patients with UC when compared to healthy people (11.4 ± 7.2 pg/mL vs. 7.2 ± 1.4 pg/mL) [115]. Conversely, the concentration of active AM in patients with severe ARDS in COVID-19 increased to five times the normal range: 101.9 {67.0–201.1} pg/mL (median and interquartile range), whereas the reference in healthy people was 20.7 pg/mL [103]. In our phase 1 trial, plasma levels of active AM were increased to 6.3 to 8.6 times of those in healthy people by repeated administration of the highest dose of AM (day1: 45.7 ± 12.7, day7: 62.0 ± 23.4 pg/mL) [114]. Our group, along with Simon. et al., are using different antibodies for the measurement of active AM, so our values tend to be lower than Simon, T.P. et al. [108]. Moreover, similar results were obtained in patients with UC in a phase 2 trial, namely the concentration of AM reached 5.9 to 7.4 times the basal values; day 1: 67.4 ± 16.0 and day 8: 84.7 ± 48.1 pg/mL [115]. Therefore, the highest dose of AM in our trial seems to be sufficient for the treatment of severe COVID-19. The first patient was enrolled in November 2020, and the enrollment was finished in December 2021, before the recent Omicron variant wave. We have started the evaluation of the data, and the results will be released in the near future.

After the start of the trial for severe pneumonia, many researchers requested early intervention for progressing pneumonia using AM to reduce the need for mechanical ventilation. Early intervention can alleviate progressive pneumonia and improve patient prognosis. Therefore, we started a second trial with an expanded subject target that included patients with moderate pneumonia who required oxygen support; in addition, we began an investigator-initiated phase 2a, randomized, double-blind, multicenter, placebo-controlled clinical trial (https://jrct.niph.go.jp/latest-detail/jRCT2071210038 (accessed on 23 June 2021)). The inclusion criterion is patients who require oxygen support for respiratory failure caused by COVID-19 within 10 days of the onset of symptoms. The primary endpoint was the duration of oxygen support. Major secondary endpoints include clinical six-category ordinary scores until 30 days after the start of the test drug, which is the same as in the trial for severe pneumonia. Continuous intravenous infusion of AM was performed for 72 h, followed by intermittent administration for 8 h per day until release from oxygen support (Figure 4B). The first patient was enrolled in this trial in July 2021, and enrollment is ongoing. Cytokines and markers for coagulation and the fibrinolytic system were also investigated in these two clinical trials. 

The expected effects or estimated mechanisms observed in experimental animals are not always shown in real patients. For example, the beneficial effects of adrecizumab in patients with septic shock or cardiogenic shock were not confirmed in phase 2 patient trials, although positive effects were confirmed in experimental models [74,116]. The advantage of AM is that a positive effect in real patients with inflammatory bowel disease was confirmed in a clinical trial [115]. Additionally, AM is an endogenous substance and does not exhibit strong immune suppression; therefore, it is safe and can be combined with other drugs, such as steroids or other immune modulators, such as baricitinib or tocilizumab.

## 8. Conclusions

AM is a crucial bioactive peptide in the maintenance of vascular function. The KO of AM or AM1 receptor causes lethal vascular abnormalities at mid-gestation. AM is markedly increased in patients with severe infections, such as sepsis or COVID-19; the increase reflects the severity of the infection and accurately predicts prognosis. Exogenous AM administration reduces organ damage due to infections and decreases the mortality rate in experimental septic models. Unfortunately, these effects have not been confirmed in patients with sepsis. However, the evidence is sufficient to expect beneficial effects in real patients. The close relationship between AM and COVID-19 has been confirmed; therefore, we started investigator-initiated clinical trials using AM to treat moderate to severe pneumonia in severe COVID-19 cases. If positive effects are confirmed in the trials, AM could be an innovative, naturally occurring treatment in severe COVID-19 that could have a significant public benefit.

## Figures and Tables

**Figure 1 biomedicines-10-00533-f001:**
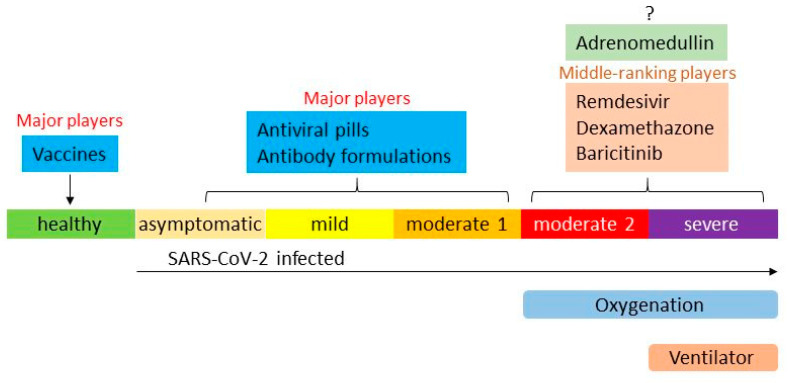
Current COVID-19 therapeutics, including adrenomedullin, under investigation in clinical trials.

**Figure 2 biomedicines-10-00533-f002:**
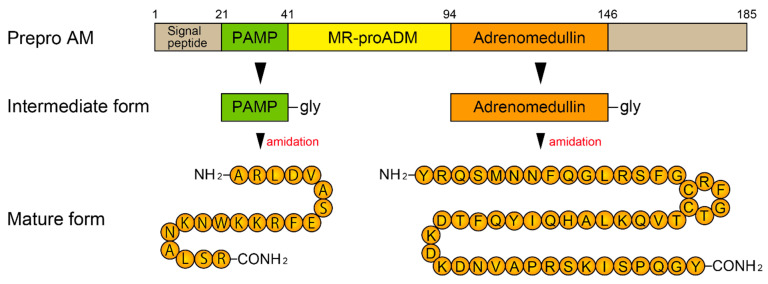
Biosynthesis of adrenomedullin. AM: adrenomedullin; PAMP: pro-adrenomedullin N-terminal 20 peptide; MR-proADM: mid-regional pro-adrenomedullin.

**Figure 3 biomedicines-10-00533-f003:**
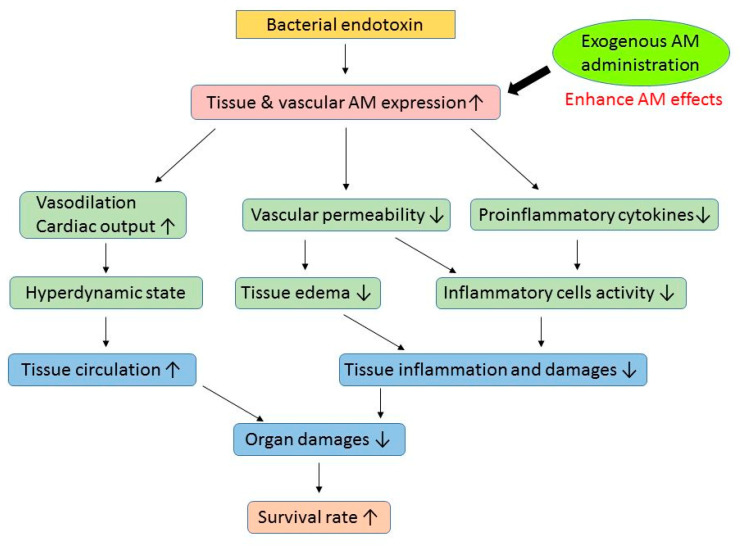
Simplified adrenomedullin cascade, an organ-protective factor against sepsis. AM: adrenomedullin.

**Figure 4 biomedicines-10-00533-f004:**
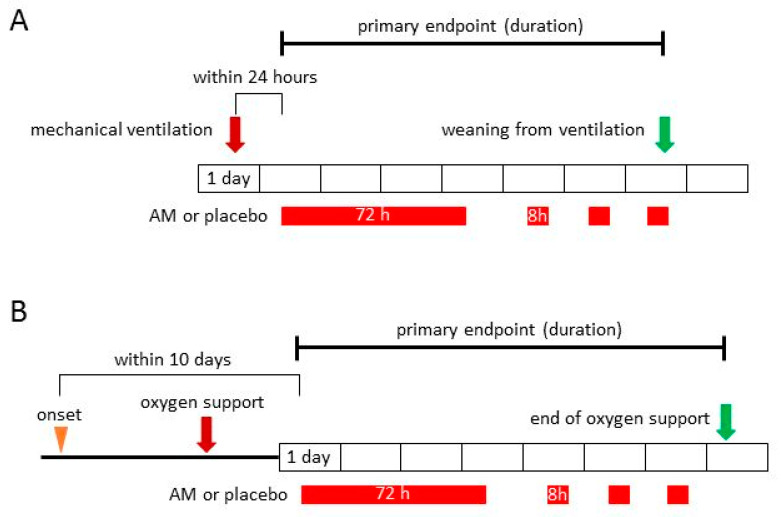
Study designs for severe pneumonia (**A**) and moderate pneumonia (**B**) in patients with COVID-19. AM: adrenomedullin.

**Table 1 biomedicines-10-00533-t001:** The effects of adrenomedullin on septic models.

**Genetic Intervention**			
**Animal**	**Procedure**	**Results**	**Reference**
Mouse	AM-deficient (+/−) + LPS-endotoxemia	compared to WT mice · ↑ mortality · ↑ liver dysfunction	[58]
Mouse	AM-deficient (+/−) + LPS-endotoxemia	compared to WT mice · ↑ TNF-α, IL-1β · ↑ liver dysfunction	[59]
Mouse	AM transgenic + LPS-endotoxemia	compared to WT mice · ↓ BP decline · ↓ organ damage · ↑ survival rate	[60]
**Exogenous Adrenomedullin Administration**			
**Animal**	**Procedure**	**Effects**	**Reference**
Mouse	Pneumococcal pneumonia + Mechanical ventilation	· ↓ VILI (pulmonary permeability↓) · ↓ liver and gut injury	[25]
Rat	BDL + CLP (obstructive jaundice + polymicrobial sepsis)	· ↓ tissue injury and inflammatory responses · ↑ survival rate	[61]
Rat	Staphylococcus aureus α-toxin induced septic shock	· ↓ translocation of dextran from the gut into the systemic circulation	[62]
Rat	Cecal ligation and puncture (CLP)	· ↓ tissue injury · ↓ proinflammatory cytokine levels · ↓ intestinal-barrier dysfunction · ↑ survival rate	[63]
Sheep	Endotoxin (LPS) infusion	· ↑ cardiac index · ↓ mean pulmonary artery pressure	[64]
Rat	Endotoxin (LPS) injection	· ↑ PPER-γ level · ↓ TNF-α	[65]
Rat	Intestinal ischemia/reperfusion	· ↓ lung injury · ↓ proinflammatory cytokines	[66]
Rat	Staphylococcus aureus α-toxin induced septic shock	· ↓ vascular hyperpermeability · ↑ survival rate	[67]
Rat	Intestinal ischemia/reperfusion	· ↓ inflammatory cytokines · ↓ tissue injury · ↑ survival rate	[68]

AM: adrenomedullin, LPS: lipopolysaccharide, WT: wild type. TNF: tumor necrosis factor, IF: interferon, BP: blood pressure. VILI: ventilator induced lung injury, BDL: common bile duct ligation. PPER: peroxisome proliferator-activated receptor.

## Data Availability

All data are provided in the review article.

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
