# Peer review of "Adrenomedullin Therapy in Moderate to Severe COVID-19"

_biomedicines, 2022, doi:10.3390/biomedicines10030533_

Round 1

Reviewer 1 Report

The electronically submitted MS “Adrenomedullin therapy in moderate to severe COVID-19” by Kita T and Kitamura K, is a review article that aims to present a state of the art regarding the potential use of adrenomedullin, an endogenous peptide that maintains vascular tone and endothelial barrier function, in individuals with moderate to severe COVID-19. The authors have prepared an exhaustive review of the literature and they also present preliminary findings of two ongoing clinical trials. The authors suggest adrenomedullin could be an alternative therapy in severe inflammation disorders, including COVID-19. The manuscript is well-written with interesting and appropriate references.

Below I leave a few comments so that the authors could take into consideration, if they wish, for a revised version of the manuscript.

  1. I believe that it is important to define what is severe and what is moderate covid-19. In the figure 1, it is getting more complicated with moderate 1 and moderate 2.
  2. AM and AM receptors are ubiquitously expressed in the body, including the heart, lungs, kidneys, adrenal glands, and intestines: Please, add references.
  3. AM is one of the strongest vasodilators among the vasoactive peptides and maintains vascular tone through direct action on vascular smooth muscle cells and nitric oxide produced in the endothelium: I think that this observation is based on animal models. Please check it and add a comment. If the whole paragraph is based on observations in animal models, I believe that it is important that it should be specified. 

An excellent study. Very well-structured manuscript. What is known and what is under research are clearly presented. English is perfect and the manuscript is easy to follow.

Author Response

  1. I believe that it is important to define what is severe and what is moderate covid-19. In the figure 1, it is getting more complicated with moderate 1 and moderate 2.

Response: We have added an explanation concerning the disease severity classification according to Japanese guidelines.

  1. AM and AM receptors are ubiquitously expressed in the body, including the heart, lungs, kidneys, adrenal glands, and intestines: Please, add references.

Response: We have added Ref. 49 (a comprehensive review of AM).

  1. AM is one of the strongest vasodilators among the vasoactive peptides and maintains vascular tone through direct action on vascular smooth muscle cells and nitric oxide produced in the endothelium: I think that this observation is based on animal models. Please check it and add a comment. If the whole paragraph is based on observations in animal models, I believe that it is important that it should be specified.

Response: Many researchers, including us, have confirmed the hypotensive effects of AM, and I can confidently say that AM is one of the most potent vasodilators in humans. We have added Ref. 51, a study that compared the hypotensive effects of AM and atrial natriuretic peptide, a commonly used therapeutic agent for heart failure, proving its strong vasodilative action. Additionally, the involvement of NO in vasodilation caused by AM has been proven in human coronary arterioles (Ref. 52).

Reviewer 2 Report

The review of Kita and Kitamura is a well-written and documented paper describing the Covid-19 pandemic, analyzing its level of criticality from the clinical point of view. In vitro and in vivo evidences are reported in order to demonstrate the usefulness and efficacy of AM, and a phase 2a clinical trial for COVID-19 seems to be able to provide promising results. In combination with adrecizumab, AM could produce a strong improvement of the disease, where other measures are not completely efficient as yet.

I wish the Authors could present the new data soon (I expect this is good news!).

In my opinion, the review is suitable for the publication.

Author Response

The review of Kita and Kitamura is a well-written and documented paper describing the Covid-19 pandemic, analyzing its level of criticality from the clinical point of view. In vitro and in vivo evidences are reported in order to demonstrate the usefulness and efficacy of AM, and a phase 2a clinical trial for COVID-19 seems to be able to provide promising results. In combination with adrecizumab, AM could produce a strong improvement of the disease, where other measures are not completely efficient as yet.

I wish the Authors could present the new data soon (I expect this is good news!).

Response: Thank you for your respectful comments.

Reviewer 3 Report

There are some comments concerning the manuscript.

1) It would be better to include some additional sections about adrenomedullin's biological actions, signalling pathways activated and innate immunity effects. In biological actions section, apart from vascular effects, other such as endocrine or renal can be included. In signal transduction section cAMP and other molecules can be mentioned.

2) It would be helpful to summarize in a table the in vitro effects of adrenomedullin (the table may include the reference, the effect, the pathway and the participating receptor).

3) In section 3 (AM and sepsis, second paragraph), AM and AM receptor expression pattern is mentioned. I believe that this expression profile could be described earlier, in another paragraph. 

Author Response

1) It would be better to include some additional sections about adrenomedullin's biological actions, signalling pathways activated and innate immunity effects. In biological actions section, apart from vascular effects, other such as endocrine or renal can be included. In signal transduction section cAMP and other molecules can be mentioned.

2) It would be helpful to summarize in a table the in vitro effects of adrenomedullin (the table may include the reference, the effect, the pathway and the participating receptor).

Response: Thank you for your thoughtful comments. In this review, we presented the reason why we initiated a clinical trial for COVID-19. The suggested information is very important but not essential for the clinical trial. To confirm, “whether AM works in real patients or not” is the only purpose of the trial. Therefore, we focused on the actual supportive data that can drive us to the trial. We can provide the suggested data in separate paper, if AM proves to be effective in patients.

3) In section 3 (AM and sepsis, second paragraph), AM and AM receptor expression pattern is mentioned. I believe that this expression profile could be described earlier, in another paragraph.

Response: We agree with this suggestion. However, to emphasize the systemic effect of AM in sepsis, we have included a sentence in the relevant section.